# NuMA-microtubule interactions are critical for spindle orientation and the morphogenesis of diverse epidermal structures

Lindsey Seldin[1,2†], Andrew Muroyama[1,2], Terry Lechler[1,2*]

[1] Department of Dermatology, Duke University Medical Center, Durham, United States; [2] Department of Cell Biology, Duke University Medical Center, Durham, United States

**Abstract** Mitotic spindle orientation is used to generate cell fate diversity and drive proper tissue morphogenesis. A complex of NuMA and dynein/dynactin is required for robust spindle orientation in a number of cell types. Previous research proposed that cortical dynein/dynactin was sufficient to generate forces on astral microtubules (MTs) to orient the spindle, with NuMA acting as a passive tether. In this study, we demonstrate that dynein/dynactin is insufficient for spindle orientation establishment in keratinocytes and that NuMA's MT-binding domain, which targets MT tips, is also required. Loss of NuMA-MT interactions in skin caused defects in spindle orientation and epidermal differentiation, leading to neonatal lethality. In addition, we show that NuMA-MT interactions are also required in adult mice for hair follicle morphogenesis and spindle orientation within the transit-amplifying cells of the matrix. Loss of spindle orientation in matrix cells results in defective differentiation of matrix-derived lineages. Our results reveal an additional and direct function of NuMA during mitotic spindle positioning, as well as a reiterative use of spindle orientation in the skin to build diverse structures.

*For correspondence: terry.lechler@duke.edu

Present address: † Department of Cell and Developmental Biology, Vanderbilt University Medical Center, Nashville, United States

Competing interests: The authors declare that no competing interests exist.

## Introduction

The development of complex tissues requires the coordination of cell fate decisions with the generation of correct tissue architecture. Asymmetric cell divisions (ACDs) serve to harmonize these two processes by both generating cellular diversity and dictating tissue structure (*Betschinger and Knoblich, 2004*; *Poulson and Lechler, 2012*). ACDs are driven by robust positioning of the mitotic spindle, a tightly regulated process that is orchestrated by several conserved proteins. However, our incomplete understanding of the mechanism underlying spindle orientation, as well as the lack of genetic tools to specifically perturb this process in adult mammalian tissues have prevented us from appreciating its function in various organ systems.

An LGN/NuMA/dynein-dynactin complex is a conserved regulator of spindle orientation in a number of cell types, including epidermal and neural progenitors in mammals, neuroblasts in *Drosophila* and zygotic divisions in *Caenorhabditis elegans*. Loss-of-function studies on these proteins have revealed their essential roles in spindle orientation and force generation (*Bowman et al., 2006*; *Izumi et al., 2006*; *Nguyen-Ngoc et al., 2007*; *Siller et al., 2006*; *Skop and White, 1998*; *Williams et al., 2011*). Furthermore, targeting dynein/dynactin to the cell cortex was sufficient to induce spindle movements, consistent with it playing a direct role in force generation (*Kotak et al., 2012*). These data have led to the proposition that LGN and NuMA form a passive tether, which

**eLife digest** Before a cell divides, it must duplicate its DNA so that each new cell receives a complete set of genetic material. A structure called the mitotic spindle helps to ensure each new cell gets the correct amount of DNA. Cells often precisely position their mitotic spindle during division, and this spindle orientation is important for generating different types of cells and for establishing the three-dimensional structure of tissues. How cells rotate their spindles into the correct position is not well understood, but a protein called NuMA is important for this process.

Seldin et al. developed genetic tools that could disrupt spindle orientation in specific types of cells to determine where this orientation is important for proper tissue development. This revealed that the correct placement of the mitotic spindle is important for the development of the skin of mouse embryos and the formation of the hair of adult mice. Seldin et al. also found that the NuMA protein binds to the tips of the microtubules that make up the mitotic spindle. This binding activity is important for NuMA to be able to position the mitotic spindle correctly in the cell. The findings suggest similarities between how cells orient mitotic spindles and how they segregate DNA during cell division.

More work is now needed to better understand how NuMA collaborates with force-generating molecular motors to precisely orient the mitotic spindle in the cell. In addition, understanding how spindle orientation dictates the fate of cells in the skin is an important future goal.

provides a high level of regulatory control over dynein accumulation at the cell cortex, and that dynein-dependent forces position the spindle (*Kotak et al., 2012*; *Kotak and Gonczy, 2013*).

The regulation of microtubule (MT) dynamics is another important factor in driving proper spindle orientation. For example, local taxol treatment at the posterior end of the *C. elegans* zygote prevented spindle movement toward that pole (*Nguyen-Ngoc et al., 2007*). Additionally, in contrast to the current model in metazoans, the spindle orientation process in yeast requires not only dynein-dependent pulling forces, but also MT depolymerization at the cell cortex (*ten Hoopen et al., 2012*). Present models include a kinetochore-like mechanism of capturing the energy from MT depolymerization to power spindle movement. Therefore, there is precedence for MT dynamics playing an important role during spindle orientation.

In addition to its ability to recruit dynein/dynactin to the cell cortex, NuMA contains a domain that directly interacts with MTs (*Du et al., 2002*; *Haren and Merdes, 2002*). This MT-binding domain (MTBD) is conserved among flies, worms, and mammals and has been characterized to stabilize and bundle MTs. Whether the MT-binding activity of NuMA is important for mitotic spindle orientation has not been tested.

The epidermis has emerged an as ideal tissue to understand the mechanism and functions of spindle orientation (*Lechler and Fuchs, 2005*; *Poulson and Lechler, 2010*; *Williams et al., 2011*). The tissue architecture allows for robust and reproducible determination of spindle angles while genetic and cell culture systems have allowed further exploration of the mechanisms. In embryonic development, spindle orientation is required for proper stratification and differentiation of the epidermis. The role of spindle orientation in epidermal appendages like hair follicles, which are highly organized structures, has not yet been reported. That said, stereotypical spindle orientations have been reported at several stages of hair follicle morphogenesis, suggesting that they may play important roles (*Niessen et al., 2013*; *Rompolas et al., 2012*). Direct testing of this will require development of genetic tools that specifically disrupt spindle orientation.

Here, we report that NuMA can specifically localize to the tips of MTs, consistent with a role in mediating cortex/astral MT interactions. Loss of NuMA's MTBD resulted in spindle orientation defects both in intact epidermis and in the hair follicle, leading to perturbation of differentiation and loss of tissue function.

## Results

### NuMA localizes to microtubule tips

Previous studies have defined NuMA's minimal MTBD (*Figure 1A*) (*Du et al., 2002*; *Haren and Merdes, 2002*). When expressed in cultured mouse keratinocytes, however, we observed only weak co-localization between GFP-MTBD and MTs (*Figure 1E*). This was true at both high and low levels of expression, and including or excluding the adjacent LGN-binding domain of NuMA (*Figure 1— figure supplement 1*). By contrast, when constructs that encoded additional amino-terminal regions were transfected, we noted robust association with the MT lattice in cells that expressed high levels of the fusion proteins (*Figure 1—figure supplement 1*). This was true in constructs containing the linker region between the 4.1 and LGN binding domains of NuMA, but not those that lacked this region. When highly overexpressed, the fusion proteins appeared to stabilize and bundle MTs (*Figure 1—figure supplement 1*). At lower levels of expression, however, we observed punctate MT localization of the fusion proteins that contained the linker region, LGN binding domain and the MTBD (*Figure 1B,C*). Zoomed-in views of these images show that these puncta localize along MTs and often accumulate at MT tips (*Figure 1B',C'*). The MT density is very high in many keratinocytes, which makes it challenging to see a precise co-localization. We therefore examined the localization of these fusion proteins in regions of the cell periphery where MTs were sparse. Co-staining with α-tubulin revealed discrete localization to MT plus tips near the cell periphery (*Figure 1G–G''*). Thus, a region of NuMA spanning the linker domain (which follows the 4.1-binding domain) to the MTBD localizes to MT tips. We refer to this entire region as 'NuMA-TIP' (*Figure 1—figure supplement 1*).

Eb1 is a well-characterized plus tip-binding protein that interacts specifically with growing MT ends. However, co-staining revealed that NuMA-TIP and Eb1 only very rarely colocalized on the same MT tip (*Figure 1H,K,K'* <2% of MTs, n = 200). This could be due to their mutually exclusive steric interactions and/or association with distinct MT end structures.

To better characterize the properties of the NuMA-TIP protein fragment, we examined how alterations in MT dynamics affected its localization. Many +TIP tracking proteins bind exclusively to growing MT ends and are displaced under depolymerization conditions, which can be triggered using nocodazole treatment. In stark contrast, NuMA-TIP localized specifically to both MT plus and minus ends following treatment with nocodazole (*Figure 1I,K',K''*). This was in contrast to Eb1, which was displaced from the tip and labeled the MT lattice upon nocodazole addition (*Figure 1K''*). Since nocodazole significantly decreased the density of MTs and helped resolve MT ends, this treatment revealed that NuMA-TIP clearly favors binding to MT ends over MT sides. This same localization pattern was also observed in transfected HeLa cells, demonstrating that this MT tip-binding behavior of NuMA is not unique to keratinocytes (*Figure 1—figure supplement 2*).

While MT plus and minus ends are structurally distinct, both undergo depolymerization that involves protofilaments curling off of the tube (*Tran et al., 1997*). NuMA-TIP localization to nocodazole-treated MT ends suggested that it might preferentially bind to depolymerizing MTs and/or to those exhibiting protofilament curling. To determine whether NuMA-TIP localization correlates with the status of MT tips, we treated cells with either low doses of vinblastine, which promotes protofilament curling, or taxol, which straightens MT ends (*Elie-Caille et al., 2007*). Under the vinblastine conditions used, this drug did not significantly alter MT density or organization. We found that a higher percentage of vinblastine-treated cells showed MT tip localization of NuMA-TIP when compared with DMSO-treated cells, while taxol treatment resulted in a significant loss of MT tip localization (*Figure 1J*). This suggests that NuMA preferentially recognizes flayed MT ends over straight ends and is consistent with a report that NuMA interacts preferentially with vinblastine-induced curled MTs over taxol-stabilized MTs (*Volkov et al., 2015*). Nevertheless, while NuMA-TIP co-localized well with MT tips in the presence of low-dose vinblastine, we did not observe association with the tubulin aggregates formed in cells treated with higher doses of vinblastine (data not shown).

Endogenous NuMA also exhibited MT tip localization. In early prometaphase, before NuMA had strongly accumulated at both the spindle poles and the cortex, we noted discrete puncta at MT tips (*Figure 2A*). Briefly treating cells with nocodazole resulted in enhanced tip localization (*Figure 2B*). Thus, MT-tip binding is a property of full-length endogenous NuMA.

While the NuMA-TIP region was sufficient for NuMA's interaction with MTs, NuMA could also potentially interact with MTs indirectly through dynein/dynactin. We therefore examined which regions of NuMA were necessary for MT tip localization. Analysis of full-length NuMA localization

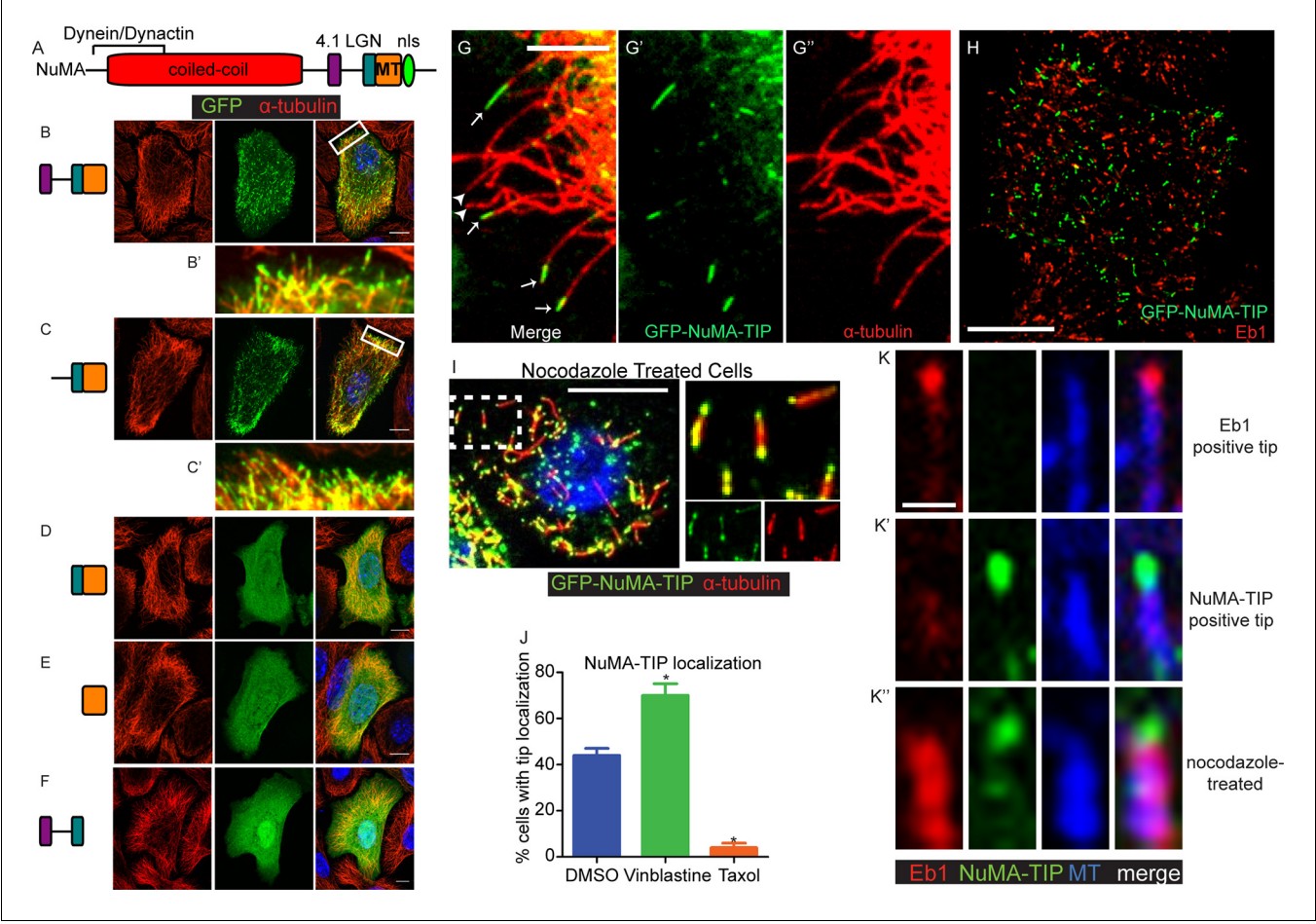

**Figure 1.** NuMA localizes to microtubule tips. (**A**) Diagram of NuMA showing interaction domains for dynein/dynactin, 4.1 family proteins, LGN, and MTs as well as the nuclear localization sequence (NLS). (**B–F**) Visualization of GFP-tagged NuMA constructs (as diagrammed) and α-tubulin (red) in cultured mouse keratinocytes. All cells displayed expressed GFP constructs at low levels. (**B'** and **C'**) are zoomed-in views displaying the punctate localization of these constructs along MTs and MT tips. (**G–G''**) Co-localization of GFP-NuMA-TIP with MT ends (arrows) at the periphery of a cultured keratinocyte. Arrowheads indicate MT tips that lack NuMA-TIP label (scale bar, 5 µm). (**H**) Keratinocytes were transfected with GFP-NuMA-TIP and then fixed and stained for Eb1 (red). Note the lack of co-localization between NuMA-TIP and Eb1 puncta. (**I**) Keratinocytes were treated with 10 µM nocodazole for 15 min and then fixed before visualizing GFP-NuMA-TIP and MTs (red). Images on right show a higher magnification view of NuMA-TIP localizing to both ends of shortened MTs. (**J**) Quantitation of the percent of cells showing GFP-NuMA-TIP localization to MT tips following drug treatments (only low-expressing cells were analyzed). Keratinocytes were treated with either DMSO, 2 nM vinblastine, or 10 µM taxol (n>100 cells each, p<0.05 for each treatment relative to control). (**K–K''**) Three-color staining for Eb1 (red), GFP-NuMA-TIP (green) and MTs (blue) in either untreated keratinocytes (top 2 panel rows) or nocodazole treated keratinocytes (bottom panels). Scale bar is 0.5 µm. Unless noted, all scale bars are 10 µm. MT, microtubule.

The following figure supplements are available for figure 1:

**Figure supplement 1.** Localization of NuMA fragments when highly expressed.

**Figure supplement 2.** HeLa cells transfected with GFP-NuMA-TIP were then treated with either DMSO (**A**) or 10 µM nocodazole (**B**), fixed and stained with anti-α-tubulin antibodies (red).

requires the examination of mitotic cells, since NuMA is sequestered in the nucleus in interphase. To circumvent this issue, we examined the localization of NuMA fusion constructs lacking the nuclear localization sequence (NLS) alone, or in combination with the loss of the MTBD or the entire NuMA-TIP domain (*Figure 2C*). We observed association of GFP-NuMAΔNLS with MT tips when the fusion protein was expressed at low levels (*Figure 2D*). Conversely, cells expressing a NuMA construct that

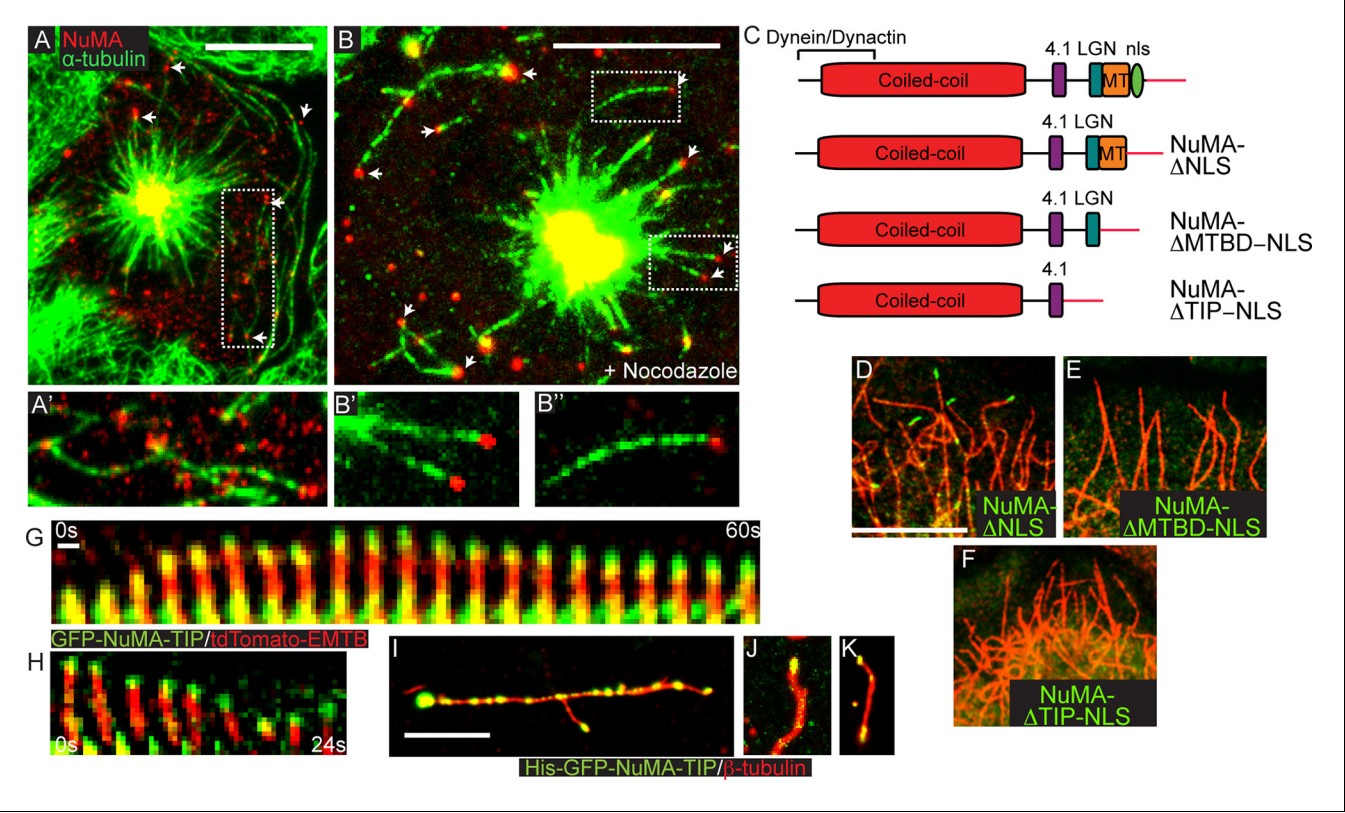

**Figure 2.** NuMA-TIP is necessary and sufficient for microtubule tip localization. (**A**) Localization of endogenous NuMA (red) at MT tips (green) in a keratinocyte in early prometaphase. (**B**) Keratinocytes were treated with 500 nM nocodazole for 5 min before fixation and analysis of NuMA (red) and MT (green) localization. Insets in **A** and **B** show high magnification images of NuMA on MT tips. Scale bars are 10 µm. (**C**) Diagrams of NuMA structure and the NuMA mutants generated. (**D–F**). Localization analysis of GFP-tagged NuMA mutants with MTs (red) as indicated. Scale bars are 10 µm. (**G,H**) Kymographs of time-lapse movies of GFP-NuMA-TIP and tdTomato-EMTB expressing keratinocytes, taken at the periphery of the cell (scale bar, 1 µm). (**I–K**) 6XHis-GFP-NuMA-TIP was purified from bacteria, mixed with polymerized MTs and pelleted onto coverslips. (**I**) Punctate labeling of MTs (red) by 6X-GFP-NuMA-TIP, (approximately 300 nM). (**J,K**) Tip binding of GFP-NuMA-TIP at lower concentrations of the protein (approximately 30 nM). Scale bars are 5 µm.

The following figure supplement is available for figure 2:

**Figure supplement 1.** Kymographs showing GFP-NuMA-TIP dynamics at the cell periphery (**A**) and at the centrosome (**B**).

lacked both the NLS and TIP region did not exhibit any MT tip localization (*Figure 2F*). This region is therefore both necessary and sufficient for NuMA's MT tip interactions. Importantly, deletion of the MTBD domain alone (which keeps the LGN-binding domain intact) compromises tip-localization (*Figure 2E*). The LGN-binding domain is required for cortical targeting of NuMA, which is necessary for its role in spindle orientation. Therefore, a NuMA mutant harboring a MTBD domain deletion allows us to test the role of MT tip-binding activity during spindle orientation without disrupting other critical domains of NuMA that are known to be required for its cortical localization.

The association of NuMA with MT ends after nocodazole treatment suggested that it might associate with stalled or depolymerizing MTs. To address this, we performed time-lapse analysis of GFP-NuMA-TIP dynamics in cultured keratinocytes either alone or in combination with ensconsin's MT-binding domain (EMTB) fused to tdTomato to label MTs. Live-imaging of cells expressing low levels of the construct revealed puncta that corresponded to MT tips. Many of the brighter puncta were stationary, consistent with NuMA-TIP's role in stabilizing MTs. In addition, we observed motile GFP puncta tracking toward the cell periphery along MT plus tips (*Figure 2G,H*; *Figure 2—figure supplement 1*). Furthermore, these puncta remained localized to the ends of depolymerizing MTs. This behavior was also evident at the centrosome (which displayed NuMA-TIP localization as well)

(*Figure 2—figure supplement 1*). These data demonstrate that NuMA can remain attached to both growing and shrinking MT ends and suggests that this may be functionally important for facilitating the interactions of astral MT tips with the cell cortex during spindle orientation.

To determine whether NuMA-TIP could directly interact with MTs and MT tips, we purified a GFP and 6XHis-tagged version of this protein from bacteria. We took advantage of the GFP tag to ask whether we could detect association of purified NuMA-TIP with MTs. As shown in *Figure 2I*, NuMA-TIP at high concentrations decorated the MT lattice, although we often observed enhanced accumulation at MT tips. At lower concentrations, however, we observed specific association with MT tips (*Figure 2J*). On short MTs, we also observed localization of GFP-NuMA-TIP on both MT ends (*Figure 2K*), consistent with our findings in cells. In all these experiments, MTs were treated briefly with nocodazole before fixation to promote their depolymerization. While these data cannot rule out roles for accessory proteins in modulating NuMA's association with MTs, they do demonstrate that NuMA-TIP is sufficient for direct interaction with MT ends.

## NuMA's MTBD is required for spindle orientation in cultured keratinocytes

To directly test whether NuMA's MTBD is required for spindle orientation, we took advantage of the robust ability of cultured keratinocytes to align their mitotic spindle with the cortical NuMA crescent (*Seldin et al., 2013*). We used NuMA-MTBD[fl/fl] mice (which harbor a floxed allele of Numa1 Exon 22) (*Silk et al., 2009*) to generate NuMAΔMTBD keratinocytes. Cre-induced recombination in these cells results in the in-frame deletion of NuMA's MTBD (*Figure 3—figure supplements 1*). As described above, this mutation causes loss of MT tip localization, but preserves the LGN-binding domain that is necessary for targeting NuMA to the cell cortex and promoting spindle orientation. To generate these cells, we treated primary keratinocytes isolated from NuMA-MTBD[fl/fl] mice with either adeno-GFP (control) or adeno-Cre virus (mutant), picked clones, and validated recombination by PCR (*Figure 3—figure supplement 2*). Loss of the MTBD did not impair localization of NuMA to either the spindle poles or cell cortex; however, it did cause a defect in spindle orientation (*Figure 3A–E*). More than 80% of control cells had spindles oriented within 30 degrees of the center of the cortical NuMA crescent (*Figure 3D*). In contrast, NuMAΔMTBD cells showed a complete randomization of division orientation (*Figure 3E*). Similar results were found by overexpressing NuMAΔMTBD-GFP in wild-type keratinocytes, demonstrating that this acts as a dominant-negative allele (*Figure 3—figure supplement 3*). These data demonstrate that NuMA-MT interactions are essential for proper mitotic spindle orientation establishment in cultured keratinocytes.

Considering that NuMA's canonical function in spindle orientation is to recruit dynein to the cell cortex, we tested whether dynein/dynactin's localization was perturbed in NuMAΔMTBD cells. We found that p150[glued] of the dynactin complex co-localized with the cortical crescents of both NuMA and NuMAΔMTBD (*Figure 3F,G*). This was true in 100% of cells examined (n = 50 from two experiments). In contrast, we recently published that knockdown of NuMA resulted in an almost complete loss of cortical dynactin (*Seldin et al., 2013*). Similar results were found upon overexpression of NuMAΔMTBD-GFP in wild-type cells (*Figure 3—figure supplement 4*). While the fusion protein localized normally and co-localized with dynein/dynactin, its overexpression resulted in spindle misorientation. These data demonstrate that cortical recruitment of dynein/dynactin is not sufficient for proper spindle orientation and that NuMA's MTBD provides an essential additional function.

## NuMA's MTBD is required for epidermal spindle orientation and differentiation

In contrast to cultured keratinocytes, mitotic spindles in situ align either parallel to the basement membrane to generate two basal progenitors, or perpendicular to it to generate one basal progenitor and one suprabasal cell that will differentiate (*Lechler and Fuchs, 2005*; *Poulson and Lechler, 2010*; *Smart, 1970*). To determine whether NuMA-MT interactions were critical for spindle orientation in vivo, we crossed the NuMA-MTBD[fl/fl] mouse line described above with Keratin 14-Cre mice (*Vasioukhin et al., 1999*), generating an in-frame deletion of the MTBD in epidermal progenitors. Due to the small change in protein size, we validated recombination by PCR analysis of isolated epidermis (*Figure 3—figure supplement 2*). We analyzed NuMA localization in embryonic tissue sections and confirmed that the mutant form localized normally to spindle poles and the cortex of

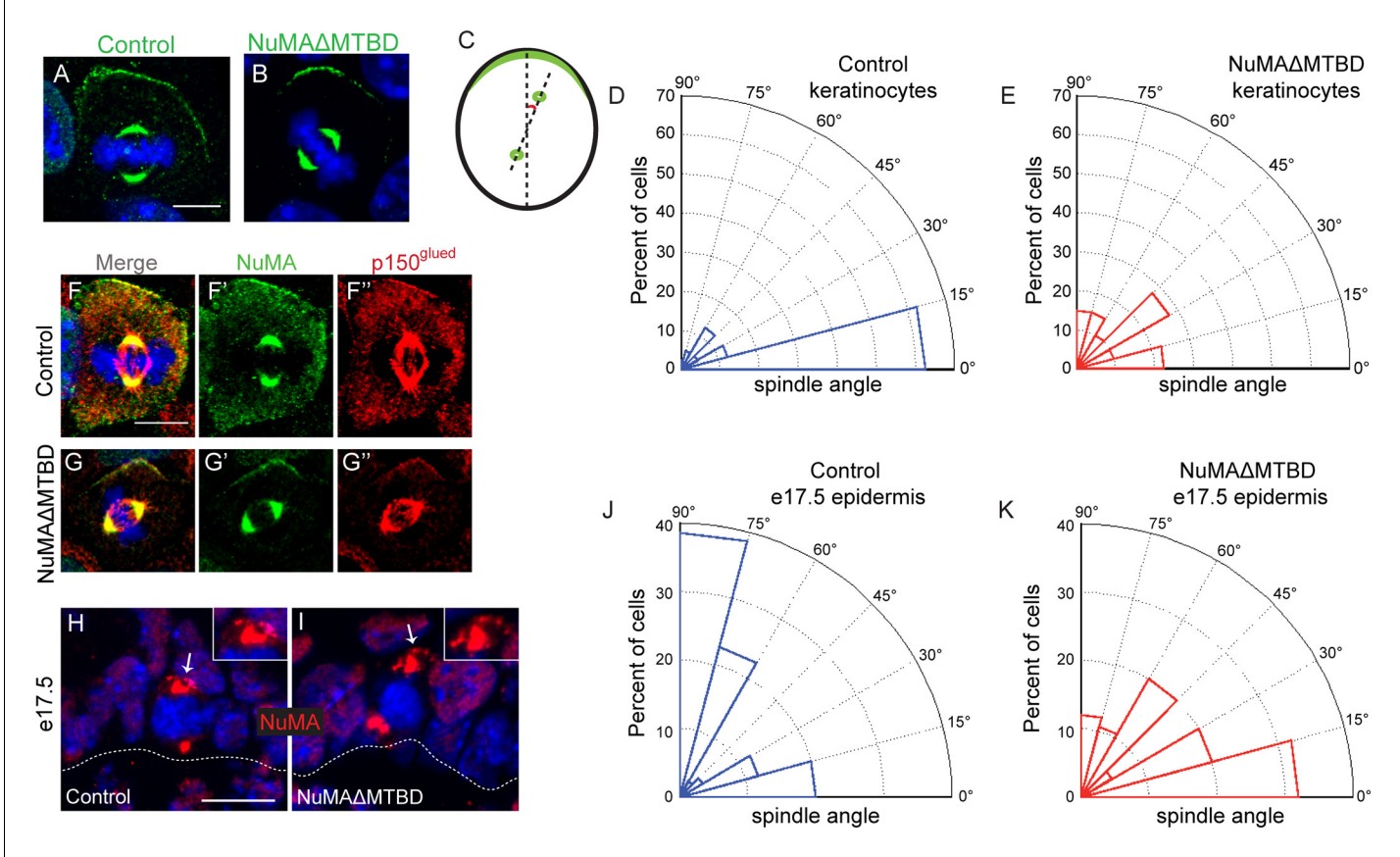

**Figure 3.** NuMA's MTBD is required for spindle orientation in cultured keratinocytes and intact skin. Localization of endogenous full-length NuMA (**A**) and NuMAΔMTBD (**B**) in mitotic keratinocytes. (**C**) Diagram illustrating how spindle angles wre measured with respect to the polarized cortical NuMA crescent in cultured keratinocytes. (**D,E**) Radial histograms representing the distribution of spindle orientation angles in control (**D**) and NuMAΔMTBD (**E**) keratinocytes (p<0.0001). (**F,G**) Co-localization of both endogenous NuMA (**F**) and NuMAΔMTBD (**G**) (green) with the p150[glued] subunit of dynactin (red) in mitotic keratinocytes. (**H,I**) Localization of NuMA (**H**) and NuMAΔMTBD (**I**) (red) in mitotic basal cells from e17.5 mouse backskin cryosections. Arrows and insets indicate the apical cortical accumulation of NuMA. Dashed lines indicate the basement membrane. Scale bars, 20 μm. (**J,K**) Radial histograms showing the distribution of spindle angles relative to the underlying basement membrane in e17.5 control epidermis (**J**) and NuMAΔMTBD epidermis (**K**) (n = 75 cells, p<0.0001). Unless noted, scale bars are 10 μm.

The following figure supplements are available for figure 3:

**Figure supplement 1.** Binding sites and exon structure of NuMA.

**Figure supplement 2.** PCR genotyping analysis of NuMAΔMTBD mice and clonal cell lines.

**Figure supplement 3.** Radial histograms of spindle orientation in cultured mouse keratinocytes expressing either full-length NuMA-GFP or NuMAΔMTBD-GFP.

**Figure supplement 4.** Co-localization of p150[glued] (a dynactin subunit) (red) with both NuMA-GFP and NuMAΔMTBD-GFP.

**Figure supplement 5.** Polarity and apoptosis markers in control and NuMA(delta symbol)MTBD epidermis.

mitotic cells, consistent with our findings in cultured cells (**Figure 3H,I**). We then scored spindle orientation using cryosections from e17.5 backskin by measuring the angle between the anaphase spindle and the basement membrane. In agreement with previous studies, controls showed a bimodal distribution of orientations (**Figure 3J**). Quantitation of spindle angles in mutant epidermis showed a dramatic decrease in perpendicular spindles coupled with an increase in oblique and parallel

spindles (*Figure 3K*). This defect was not an indirect effect of loss of polarity or cell death, as both the Golgi and centrosomes showed proper polarization and cleaved caspase 3-positive cells did not significantly increase in number (*Figure 3—figure supplement 5*). While previous work on spindle orientation in NuMA knockdown mice revealed a dramatic shift toward parallel spindles (*Williams et al., 2011*), it is important to note that the basal cells of those mice were largely devoid of NuMA, thus preventing cortical dynein recruitment. The randomized spindle angles observed in NuMAΔMTBD mice are likely due to the maintenance of cortical dynein, which cannot efficiently orient the spindle in the absence of NuMA-MT interactions.

Loss of NuMA does not cause mitotic spindle assembly defects in cultured keratinocytes, mammalian epidermis or fly neuroblasts (*Bowman et al., 2006*; *Seldin et al., 2013*; *Siller et al., 2006*; *Williams et al., 2011*). However, previous analysis of NuMAΔMTBD mice harboring a full body deletion suggested that this region was essential for development and that cultured mutant fibroblasts had mitotic spindle defects (*Silk et al., 2009*). To reconcile these findings, we analyzed mitotic spindles in both NuMAΔMTBD cultured keratinocytes and mouse epidermis. We did not detect any defects in mitotic morphology, stage, or number in mutant keratinocytes (*Figure 4A–G*). Metaphase and anaphase spindles exhibited normal morphology (*Figure 4A,B*). The localization of NuMA to the cell cortex, the incidence of abnormal spindles and the extent of mis-aligned DNA were all similar between control and NuMAΔMTBD cells (*Figure 4C–F*). This was true both in primary keratinocytes directly isolated from newborn backskin as well as the clonal lines described above. There were also no significant differences in the percentage of cells in different mitotic stages or their spindle lengths (*Figure 4F,G*). Furthermore, flow cytometry analysis of mutant epidermal cells revealed no evidence of aneuploidy or polyploidy (*Figure 4I,J*). Finally, the phenotype that we observed is distinct from those caused by mutations in mitotic regulators, arguing against an underlying mitotic defect (*Foijer et al., 2013*). This stands in stark contrast to past findings in cultured fibroblasts and thus suggests the possibility of cell-type specific roles for NuMA. Nevertheless, it should be noted that our data are consistent with previous studies on cultured keratinocytes, mouse epidermis and fly neuroblasts (*Bowman et al., 2006*; *Seldin et al., 2013*; *Siller et al., 2006*; *Williams et al., 2011*).

Considering that spindle orientation defects could result from compromised astral MT integrity, we also quantified the number of astral MTs in both control and NuMAΔMTBD mice and found no significant differences (*Figure 4H*). These data are consistent with loss of NuMA's MTBD impairing its interaction with astral MTs at the cortex and not spindle assembly.

Having established that loss of NuMA's MTBD causes spindle orientation defects but not spindle assembly defects, we sought to determine the resulting consequences on epidermal development. Mice with epidermal loss of NuMA's MTBD were neonatal lethal (*Figure 5—figure supplement 1*). This lethality was associated with a mild barrier defect, as observed by an X-gal penetration assay (*Figure 5A*). These observations led us to investigate whether underlying differentiation defects contributed to lethality. In control mice, keratin 5/14 marked a monolayer of proliferative basal cells, while keratin 10 was present in their differentiated progeny (*Figure 5B*). In mutant mice, we observed an expansion of K5/14- positive cells into suprabasal layers and their mixing with K10-positive cells (*Figure 5C*). These defects were noticed as early as e16.5, before epidermal barrier activity is fully established (*Figure 5—figure supplement 2*). Later differentiation markers, such as loricrin and filaggrin, were present at much lower levels in mutant skin, explaining in part the compromised barrier and neonatal lethality (*Figure 5D–F*). There was also a dramatic upregulation of the stress marker keratin 6 in mutant skin (*Figure 5G–I*). Consistent with a differentiation defect, we observed increased numbers of cycling cells in the mutant suprabasal epidermis by both BrdU and phospho-histone H3 staining (*Figure 5J–P*). In addition, we noted an increase in basal cell proliferation, which is likely a consequence of barrier dysfunction. In support of this, analysis of e16.5 epidermis (a stage at which the barrier is just beginning to form) revealed an increase in suprabasal cell divisions, but not in basal divisions (*Figure 5N*). These studies demonstrate that high levels of spindle misorientation are not tolerated and that the oblique divisions observed cannot promote proper differentiation of the epidermis. Previous data suggested that low levels of spindle misorientation were not detrimental to epidermal development (*Williams et al., 2014*). Therefore, there appears to be a critical threshold at which a decrease in functionally asymmetric divisions impairs development.

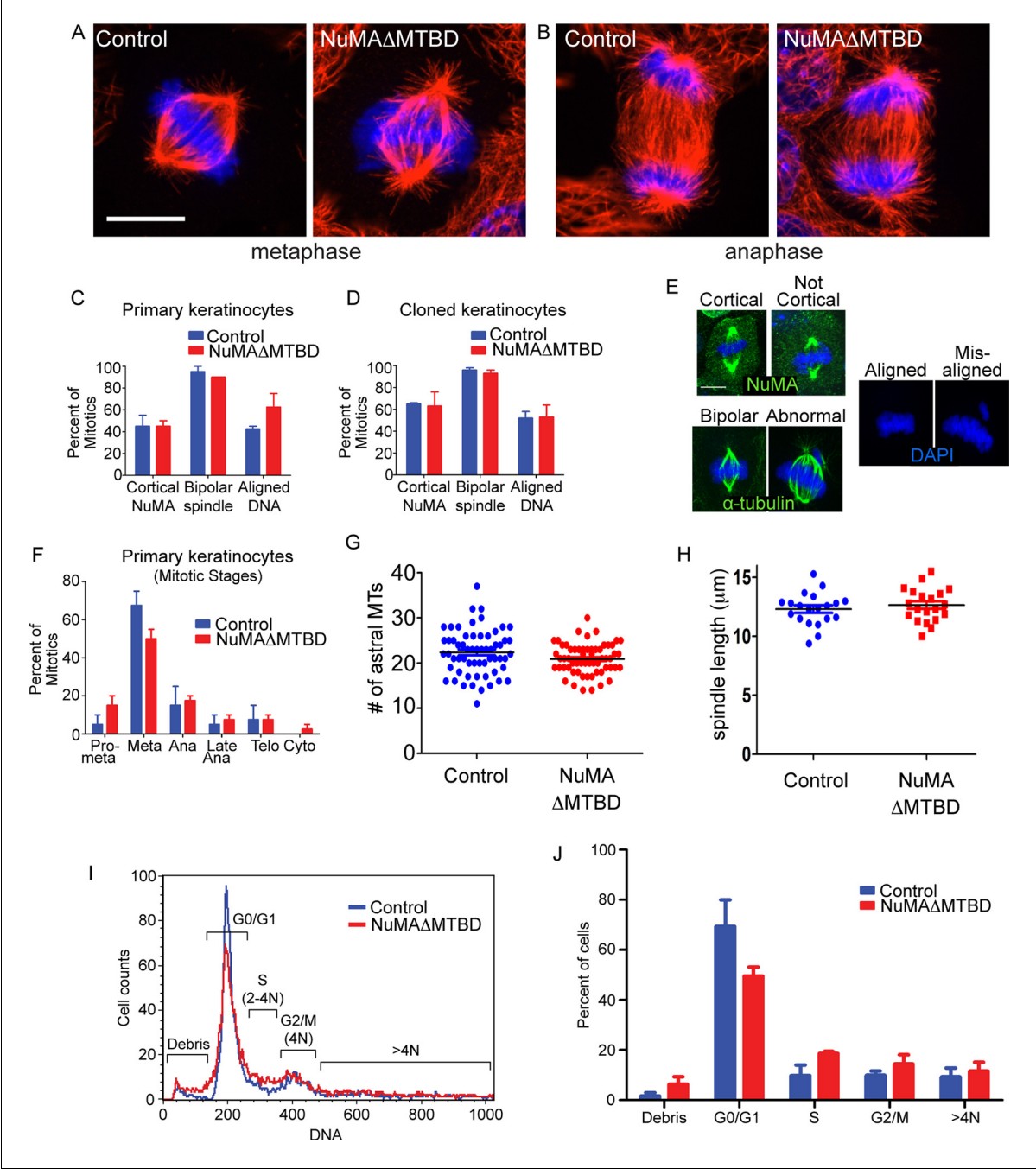

**Figure 4.** Keratinocytes show no signs of mitotic spindle assembly defects upon loss of NuMA's MTBD. (**A,B**) Immunofluorescence images of metaphase and anaphase mitotic figures from control and NuMAΔMTBD keratinocytes. MTs (red), DNA (blue). (**C**) Primary keratinocytes were isolated from control and NuMAΔMTBD mice and analyzed for cortical localization of NuMA, formation of bipolar spindles and the presence of aligned chromosomes in metaphase. n = 40, p>0.05. (**D**) Analysis as described in (**C**) was performed on control and NuMAΔMTBD clonal cell lines. n = 100, p>0.05. (**E**) Representative images of the phenotypes scored in (**C**) and (**D**). (**F**) Analysis of mitotic stages in primary cells isolated from control and NuMAΔMTBD epidermis. n = 40, p>0.05 for all. (**G**) Quantitation of the number of astral MTs/spindle pole in control and NuMAΔMTBD keratinocytes. n = 60 spindle poles, p>0.05. (**H**) Quantitation of metaphase spindle length in control and NuMAΔMTBD keratinocytes. n = 20 spindles, p>0.05. (**I**) Keratinocytes were isolated from the backskin of control and NuMAΔMTBD mice, stained with propidium iodide and their DNA content analyzed by flow cytometry. n = 2. (**J**) Bar graph of data from (**J**) representing the percentage of cells in each cell cycle stage. n = 2 mice/genotype, p>0.05 for all. All scale bars are 10 µm.

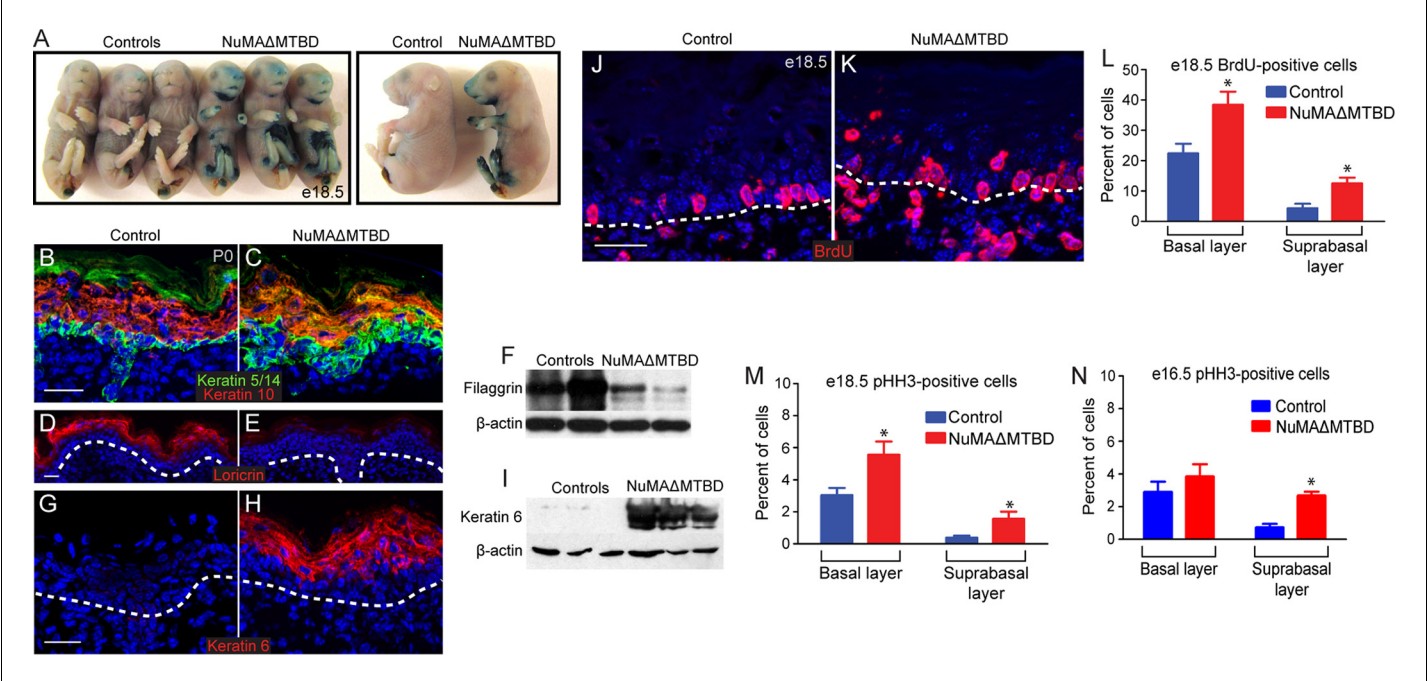

**Figure 5.** Loss of NuMA's MTBD in the embryonic epidermis results in differentiation defects and neonatal lethality. (**A**) X-gal penetration assay of e18.5 NuMAΔMTBD and littermate control mice shows a mild barrier defect in the mutant. (**B,C**) Immunofluorescence analysis of the basal cell marker keratin 5/14 (green) and the differentiated cell marker keratin 10 (red) in control (**B**) and NuMAΔMTBD (**C**) P0 backskin. (**D,E**) Expression of the granular cell marker loricrin (red) in control (**D**) and NuMAΔMTBD (**E**) epidermis. (**F**) Western blot of filaggrin levels in protein lysates prepared from control and NuMAΔMTBD epidermis. (**G,H**) Expression of the stress marker keratin 6 (red) in control (**G**) and NuMAΔMTBD (**H**) epidermis. (**I**) Western blot of keratin 6 levels in lysates prepared from control and NuMAΔMTBD epidermis. (**J,K**) Dams with e18.5 embryos were given a pulse of BrdU one hour before sacrifice. Control (**J**) and NuMAΔMTBD (**K**) embryonic backskin cryosections were stained with anti-BrdU antibodies (red) to assess proliferation rates. (**L**) Quantitation of BrdU incorporation in basal and suprabasal cells of the control and mutant epidermis (n = 3 mice/genotype, >500 cells analyzed. basal, p = 0.04; suprabasal, p = 0.03). (**M,N**) Quantitation of the percentage of phospho-histone H3 positive keratinocytes in basal and suprabasal cells layers at e18.5 (**M**) and e16.5 (n = 3 mice/genotype, >500 cells analyzed. basal, p = 0.05; suprabasal, p = 0.07). * denotes p values ≤0.05. All scale bars are 50 μm.

The following figure supplements are available for figure 5:

**Figure supplement 1.** Photographs of neonatal control and NuMAΔMTBD mice.

**Figure supplement 2.** Analysis of differentiation in e16.5 control and NuMAΔMTBD embryos.

## NuMA's MTBD is required for hair follicle spindle orientation and morphogenesis

Loss of NuMA's MTBD results in spindle orientation defects without affecting spindle assembly or cell polarity. The NuMAΔMTBD mouse line therefore serves as an invaluable tool that allows us to specifically address the role of spindle orientation in a tissue context. Previously, it has not been possible to specifically perturb spindle orientation during postnatal hair morphogenesis. The hair follicle is a highly organized epidermal appendage that undergoes cycles of growth, regression and rest throughout an animal's lifetime. The growth phase, called anagen, is responsible for producing the fully differentiated hair shaft. Three proliferative compartments of the hair follicle, the bulge stem cells, the outer root sheath (ORS) and the transit-amplifying matrix cells, are responsible for generating the cell types that make up the mature hair follicle.

We generated inducible epidermal NuMAΔMTBD mice by crossing to a keratin 5-Cre[ER] mouse line (*Figure 6A*) (*Van Keymeulen et al., 2011*). Mice were injected with tamoxifen at both P18 and P20, which coincide with the first telogen (the resting phase of the hair cycle), and a small region of backskin was shaved to observe hair regrowth in the subsequent anagen (*Figure 6B*). In contrast to

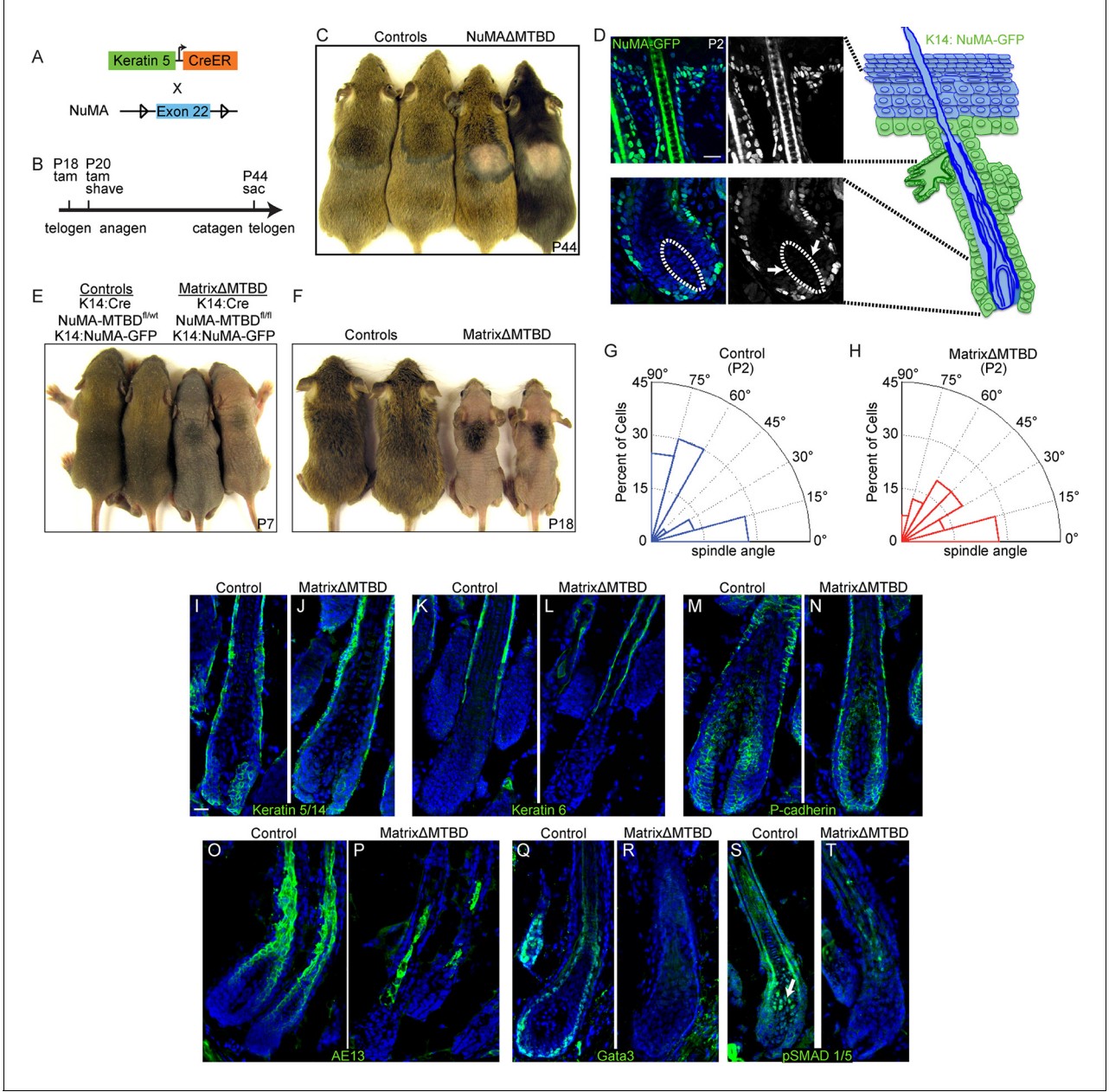

**Figure 6.** NuMA's MTBD is required in hair follicle matrix cells for proper spindle orientation, differentiation, and hair generation. (**A**) Diagram illustrating the mouse mating used to generate conditional inducible deletion of NuMA's MTBD in the adult interfollicular epidermis and hair follicles. (**B**) Diagram illustrating the experimental time course. Mice were injected twice with tamoxifen in telogen of the hair cycle, shaved and then examined for subsequent hair growth in the following anagen. (**C**) Images of control and NuMAΔMTBD mice treated as described in (**B**) and photographed at postnatal day 44. (**D**) Expression of keratin 14 promoter-driven NuMA-GFP in both the interfollicular epidermis and outer root sheath (ORS) of a P2 hair follicle. The hair shaft signal in the top images is due to autofluorescence. The dashed line indicates the basement membrane that separates the dermal papilla and the matrix cells. Note the lack of NuMA-GFP expression in the matrix cells (arrows). (**E,F**) Photographs of control and matrixΔMTBD mice (K14-Cre; NuMA MTBD$^{fl/fl}$;K14-NuMA-GFP) at postnatal day 7 (**E**) and 18 (**F**). Note that the small patch of dorsal hair is likely due to incomplete recombination and is lost in subsequent hair cycles (*Figure 6—figure supplement 4*). (**G,H**) Radial histograms of spindle angles in the matrix cells of control (**G**) and matrixΔMTBD (**H**) mice at P2 (n = 40 cells. p = 0.005). (**I–T**) Immunofluorescence analysis of the indicated hair differentiation markers (green) in control and matrixΔMTBD backskin at P4. Keratin 5/14 labels the ORS, keratin 6 labels the companion layer, P-cadherin labels matrix cells, AE13 labels the cuticle and cortex of the hair shaft, and GATA3 labels the inner root sheath. All scale bars are 20 μm.

The following figure supplements are available for figure 6:

**Figure supplement 1.** MatrixΔMTBD mice (4 months old) showing loss of all hair as the animals age.

*Figure 6 continued on next page*

*Figure 6 continued*

**Figure supplement 2.** Representative images of anaphase spindles in the matrix of the hair follicle.

**Figure supplement 3.** Hair follicle histology at P18 (**A,B**) and P30 (**C,D**) in control and MatrixΔMTBD mice.

**Figure supplement 4.** Analysis of BrdU positive cells in control and matrixΔMTBD mice at postnatal day 2.

the robust hair regrowth seen in littermates, we observed an almost complete absence of hair regrowth in the mutant mice (*Figure 6C*).

We next wanted to isolate a single proliferative compartment within the hair follicle to examine the specific effects of deleting NuMA's MTBD. Matrix cells are rapidly dividing and give rise to several distinct lineages, making them ideal candidates for utilizing spindle orientation. While bulge and ORS cells are contiguous with the interfollicular basal layer and express keratin 14, matrix cells are keratin 14-negative. Using previously generated mice in which NuMA-GFP is expressed under the control of the keratin 14 promoter (*Poulson and Lechler, 2010*), we could rescue NuMA expression in the bulge and ORS (in Krt14-Cre;NuMA-MTBD^fl/fl mice), resulting in mice with a matrix-specific MTBD deletion that we will refer to as 'matrixΔMTBD' (*Figure 6D*). Krt14-NuMA-GFP expression was sufficient to fully rescue the neonatal lethality caused by embryonic loss of NuMA's MTBD (*Figure 6E*). However, these mice displayed severe hair growth defects, indicating a critical role for NuMA's MTBD in matrix cells (*Figure 6F*). The small tufts of backskin hair present at early stages in matrixΔMTBD mice was lost during subsequent hair cycles (*Figure 6—figure supplement 1*). While these findings implicate spindle orientation in matrix function, they do not rule out the possibility that NuMA's MTBD plays a critical role within other hair cell types as well.

To determine whether spindle orientation was disrupted in matrixΔMTBD mice, we analyzed cryosections from P2 control and mutant follicular epidermis before overt changes in follicle architecture became apparent. Spindle orientation was quantitated similar to that in the embryonic epidermis by measuring the angle formed by the anaphase spindle relative to the underlying basement membrane (*Figure 6—figure supplement 2*). Compared with controls, the spindles of matrix cells lacking NuMA's MTBD exhibited a randomization of orientations (similar to the orientation pattern seen in knockout interfollicular epidermis), demonstrating that the MTBD is critical for robust spindle positioning within this cellular compartment (*Figure 6G,H*).

To determine the impact of this defective spindle orientation, we examined markers of the distinct cell lineages of the hair follicle. Keratins 5 and 6 mark the ORS and companion layer of the hair follicle, respectively, and demonstrated similar staining patterns in control and mutant follicles. Additionally, the matrix marker P-cadherin showed normal staining in both control and mutants, indicating that the matrix population was properly specified (*Figure 6I–N*). In contrast, AE13 and GATA3, which mark the hair shaft and inner root sheath (IRS) cells (all matrix descendants), respectively, were decreased and/or disorganized in the mutant follicles (*Figure 6O–R*). Therefore, robust spindle orientation is necessary for the proper differentiation of matrix progeny.

Notably, the matrix-derived hair shaft and IRS cell lineages are also lost when BMP signaling is inhibited (*Andl et al., 2004*; *Kobielak et al., 2003*; *Kulessa et al., 2000*). In fact, BMP signaling is one of the earliest detectable steps in the differentiation of these lineages. We therefore analyzed BMP activation status to determine whether spindle orientation was required for BMP activity. Analysis of BMP activity using anti-pSMAD1/5 antibodies revealed a loss of signaling (*Figure 6S,T*). These data are consistent with BMP signaling acting downstream of these asymmetric divisions. Examination of older mice revealed additional phenotypic similarities with mice in which BMP signaling was blocked, including cyst formation and aberrant spatial localization of follicular proliferation (*Figure 6—figure supplement 3*). Although matrix-derived cells in the mutant mice lacked markers for IRS and hair shaft, they appeared to be non-proliferative, as judged by lack of BrdU incorporation (*Figure 6—figure supplement 4*). One of the simplest models suggests that BMP inhibitors in the dermal papilla locally impede BMP signaling in matrix cells, while more distant cells undergo active signaling. Our data demonstrate that a spindle orientation-dependent event is also required to activate BMP signaling, which is therefore not simply driven by compartmentalization.

## Discussion

While precise control of spindle orientation has been documented in an increasing number of developmental and physiological contexts, the underlying mechanisms are only partially understood. Here, we have documented a new role for NuMA, one of the core components of the spindle orientation machinery. In addition to being important for the recruitment of dynein/dynactin to the cortex, we have found that NuMA's dynein-independent interactions with MTs are also essential for spindle orientation.

We report an unusual and previously unreported co-localization of NuMA with MT tips and define a region within NuMA that is both necessary and sufficient for this localization. While it is likely that the direct interactions provided by the NuMA-TIP region contribute to the specific binding of NuMA to MT tips, we cannot rule out that additional proteins are required to increase NuMA's MT tip-binding affinity. The properties of NuMA-TIP are distinct from many tip-binding proteins. Most notably, NuMA-TIP remains associated with stalled and or depolymerizing MTs. To date, there are very few proteins known to exhibit MT depolymerization-tracking behavior (either directly or indirectly). The few known examples are kinetochore proteins of different species, including the Dam1 complex, CENP-E, CENP-F, and the Ndc-80 complex. Of these, only Dam1 has been shown to follow depolymerizing MTs in vivo, while the others can do this in vitro, yet remain at the kinetochore-MT interface in vivo (*Westermann et al., 2006*). Depolymerizing kinesins have also been observed to localize to both plus and minus ends of MTs (*Desai et al., 1999*).

While a previous study demonstrated that artificially targeting dynein to the cortex was sufficient to induce spindle oscillations (*Kotak et al., 2012*), our data suggest that cortical dynein is not sufficient to robustly position the spindle and that NuMA-MT interactions are also required (*Figure 7*). A simple model in which all forces during spindle orientation are generated by dynein-dependent pulling on astral MTs is still formally possible, with NuMA acting simply to provide MT stabilization, thus making astral MTs better substrates for dynein-directed forces. While a viable possibility, this is not consistent with local stabilization of MTs inhibiting spindle movements (*Nguyen-Ngoc et al., 2007*). Alternatively, NuMA's unique MT-binding qualities revealed in this study could allow it to interact with depolymerizing MTs and harness their energy to promote spindle rotation, thus collaborating with dynein pulling forces to robustly orient the spindle. This latter idea is similar to kinetochores harnessing MT depolymerization for chromosome segregation (*McIntosh et al., 2010*). Both the association of MT dynamics with force generation observed during spindle positioning (*Labbe et al., 2003*), as well as work showing the ability of depolymerizing MTs to generate force (*McIntosh et al., 2010*) are consistent with this second possibility. This is not without precedent, as positioning of the budding yeast spindle relies on regulated depolymerization at the cell cortex (*Gupta et al., 2006*). If and how depolymerization activity is coordinated with dynein/dynactin will require further investigation. It is also possible that both stabilization and tracking depolymerizing MTs are important activities of NuMA that are used at distinct sites/times. For example, NuMA has been shown to collect at the minus ends of severed MTs in the mitotic spindle where it may act to stabilize the free ends (*Elting et al., 2014*).

Our data further demonstrate the developmental roles for spindle orientation in two distinct cellular environments. Consistent with previous studies, loss of spindle orientation in the developing epidermis resulted in embryonic lethality that is likely due to epidermal differentiation defects. These data raise the question of whether the early embryonic lethality observed in the full mouse NuMA MTBD knockout was due to mitotic defects, defects in spindle positioning or a combination of these effects (*Silk et al. 2009*). While the phenotype that resulted from the epidermal-specific loss of NuMA's MTBD in this study was less severe than that seen upon shRNA depletion of either LGN or NuMA, there are two plausible explanations to reconcile this discrepancy. First, the gene ablations in the previous study were initiated prior to epidermal stratification, whereas our knockout occured during the process of stratification. Second, loss of full-length NuMA or LGN resulted in a drastic conversion of ACDs (perpendicular spindles) to SCDs (parallel), while loss of NuMA's MTBD resulted in a conversion of ACDs to randomized division orientations.

The finding that the NuMA MTBD deletion primarily affects spindle orientation without observable effects on either cell polarity or cell division establishes the NuMA-MTBD deletion mice as a valuable tool to study the in vivoroles of spindle orientation in intact tissues. We have taken advantage of this by being the first to demonstrate an essential requirement for spindle orientation in hair

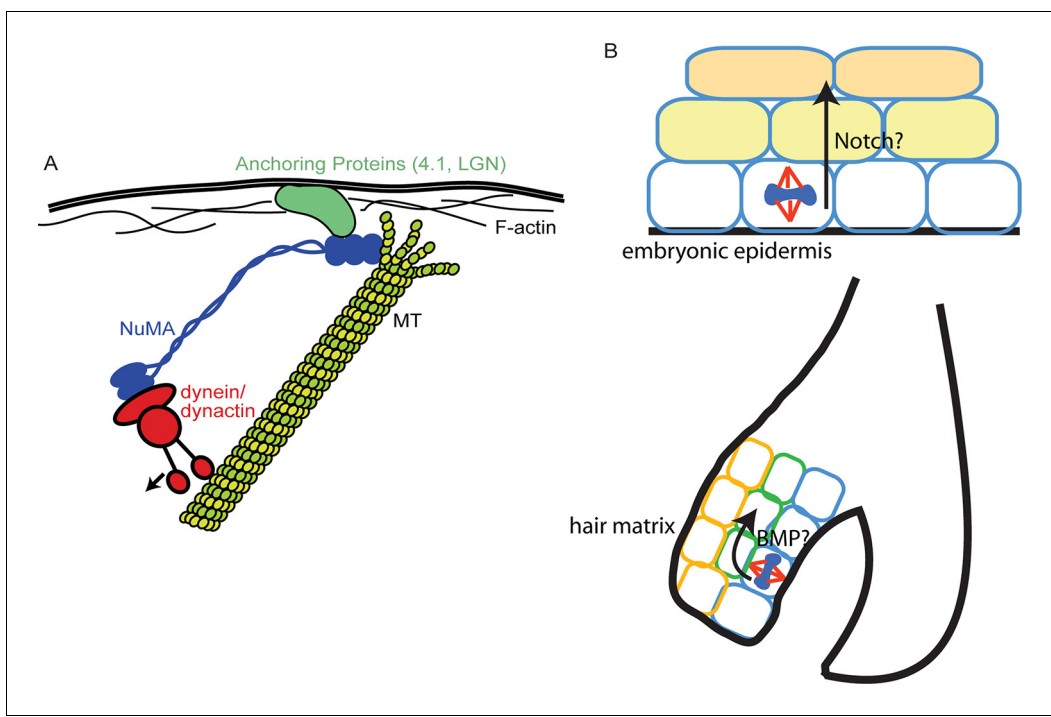

**Figure 7.** Models for NuMA and ACD functions. (**A**) Diagram of interactions required for spindle orientation. NuMA interacts with MTs via a carboxy-terminal domain, as well as with dynein/dynactin in its amino terminus. Both these interactions are likely required for robust spindle orientation. Only a single NuMA dimer is illustrated for simplicity, although it is also thought to multimerize. NuMA may act to stabilize the MT end or, alternatively, may couple its depolymerization to force generation for spindle positioning. (**B**) Comparison of the repetitive uses of spindle orientation in the embryonic epidermis and adult hair follicle. In both cases, spindles align perpendicular to the underlying basement membrane in a significant portion of cells in a NuMA-dependent manner. Whereas cell fate decisions in embryonic epidermis are Notch-dependent, cell fate choices in the hair follicle are BMP-dependent. In neither case do we understand the molecular connection between spindle orientation and the activation of these pathways.

follicle morphogenesis. While prior reports have presented correlative data to suggest that spindle orientation contributes to hair follicle morphogenesis (*Niessen et al., 2013*; *Rompolas et al., 2012*), we have quantitated division orientations within the rapidly proliferating matrix compartment and demonstrated the importance of spindle orientation in these cells. Similar to the interfollicular epidermis in the embryo, divisions can be either parallel or perpendicular to the underlying basement membrane (*Figure 7*). The percentage of matrix cell perpendicular divisions decreased in the mutant and resulted in differentiation defects in matrix-derived lineages. One of the earliest known steps of differentiation along these pathways is BMP activation (*Andl et al., 2004*; *Kobielak et al., 2003*; *Kulessa et al., 2000*), and loss of BMP signaling resulted in phenotypes similar to loss of spindle orientation. In both cases, IRS and hair shaft markers are decreased or absent. The matrix-derived cells in the mutant mice described here as well as the BMPRIA mutant are non-proliferative, suggesting that they may be segregated from pro-proliferative signals from the basement membrane and/or dermal papilla. Surprisingly, in the absence of proper spindle orientation, the activation of BMP signaling does not occur. While the compartmentalized activation of BMP could have been due solely to the presence of BMP inhibitors in the DP preventing signaling in adjacent matrix cells, our data demonstrate that a spindle orientation-dependent step is required for cells to become responsive to BMP signaling. Whether this acts directly on the BMP signaling pathway or lies upstream of it is not currently known. Understanding the molecular basis of this requirement is a clear future goal to understand how differentiation and morphogenesis are controlled. In summary, we reveal that a reiterative use of spindle orientation machinery generates very distinct structures (stratified interfollicular epithelium versus hair follicles) within the same tissue.

## Materials and methods

### Mouse lines

NuMA-MTBD$^{fl/wt}$ mice, K14-Cre, K5-CreER and K14-NuMA-GFP mice were previously reported (*Poulson and Lechler, 2010*; *Silk et al., 2009*; *Van Keymeulen et al., 2011*; *Vasioukhin et al., 1999*). All mouse experiments were performed with approval from the Duke Institutional Animal Care and Use Committee. For BrdU experiments, mice were injected with 10 mg/kg of BrdU (Sigma-Aldrich, St. Louis, MO) and were left for 1 hr before sacrifice.

### NuMAΔMTBD cell clonal analysis

Adeno-GFP and Adeno-Cre-GFP viruses (Baylor Vector Development Labs, Houston, TX) were used separately to infect NuMA-MTBD$^{fl/fl}$ mouse keratinocytes. Single cells from each infection plate were then seeded into 96-well plates, grown to confluency and subsequently trypsinized, replated and genotyped to identify knockout clones.

### Cell culture

Mouse keratinocytes were isolated from the backskins of newborn or embryonic mice. Mouse keratinocytes were grown in E low calcium medium and maintained in a 37°C incubator with 7.5% $CO_2$. Primary cells were isolated from mouse backskin by incubating the backskin overnight in a mixture of 1X phosphate-buffered saline (PBS) and Dispase II (Hoffman-La Roche, Basel, Switzerland) at a 1:1 ratio. The epidermis was then removed from the dermis and placed in a mixture of trypsin and versene at a 1:1 ratio for 3 min. Cells were resuspended in fresh media, filtered, pelleted, and plated onto glass coverslips coated with 100 μM laminin (Invitrogen, Waltham, MA), or onto fibroblast feeders for long-term passage. For drug treatments, 10 μM nocodazole (Sigma-Aldrich, St. Louis, MO) and the corresponding concentration of DMSO was incubated with cells for 30 min before fixation. In mitotic cells, in which MTs are much more dynamic, nocodazole treatment was with 0.5 μM for 5 min. Vinblastine (2 nM) and taxol (10 μM) were both from Sigma, and treatments were for 5–10 min. For spindle orientation experiments on cultured keratinocytes, cells were plated on glass coverslips coated with 100 μM laminin (Invitrogen). HeLa cells were grown at 37°C with 7.5% $CO_2$ in DMEM media containing 10% Fetal Bovine Serum with antibiotics.

### DNA content analysis

Primary control and MTBD knockout cells were isolated as described above. After filtering and centrifugation, cells were resuspended in a mixture of PBS and 70% ethanol (1:5 ratio) and placed on ice for 2 hr. Cells were then centrifuged for 5 min at 1000 rpm and the ethanol solution was removed. After an additional wash in PBS, the cell pellet was suspended in a 1 ml solution containing 0.1% Triton X-100 (EMD Millipore, Darmstadt, Germany) in PBS, 2 mg DNase-free RNase A (Qiagen, Venlo, Netherlands) and 200 μl of 1 mg/ml Propidium Iodide (Sigma-Aldrich) and incubated at room temperature for 30 min. Cell cycle analysis was then performed on these samples using a FACSCalibur analyzer (BD Biosciences, Franklin Lakes, NJ).

### Immunofluorescence staining and analysis

Cells were fixed for 3 min in −20°C methanol or 8 min in 20°C 4% paraformaldehyde. Primary antibodies used in this study include rabbit α-NuMA and rat α-BrdU (Abcam, Cambridge, MA), mouse α-p150$^{glued}$ and rat α-β4 integrin (BD Biosciences, San Jose, CA), mouse α-pankeratin (AE13), mouse α-Gata 3 and rat α-α tubulin (all from Santa Cruz Biotechnology, Dallas, TX), m α-β-tubulin (Sigma), rabbit α-keratin 6 (Covance), chicken α-keratin 5/14 (lab generated), rabbit α-phospho histone H3 and rabbit anti-pSMAD1/5 (Cell Signaling Technology, Danvers, MA), and rabbit α-activated caspase 3 (R&D Systems, Minneapolis, MN). Images were collected using a Zeiss motorized Axio Imager Z1 fluorescence microscope with Apotome attachment, an AxioCam MRm camera, a 10x, 20x, 40x, and 63×/1.4 numerical aperture (NA) Plan Apochromat objective, Zeiss Immersol 518F oil, and AxioVision Digital Image Processing Software. Photoshop (Adobe) and ImageJ software were used for postacquisition processing.

## Time-lapse imaging

Keratinocytes were plated on MatTek glass-bottom dishes (no. 1.5) coated with 100 µM laminin (Invitrogen) and transfected with GFP-tagged NuMA constructs. Time-lapse imaging was performed using a Leica DMI6000 inverted microscope, a 63×/1.4 NA Plan Apochromat objective, Leica immersion oil, and an OrcaER camera (Hamamatsu Photonics). The imaging chamber was maintained at 37°C with 5% $CO_2$. Simple PCI software was used for image acquisition (eCommerce Solutions), and Photoshop and ImageJ software were used for post-acquisition processing.

## NuMA-TIP/MT in vitro binding assay

The entire GFP-NuMA-TIP coding sequence was inserted in frame in the 6XHis tagging vector pet28. The protein was expressed in bacteria and purified over Nickel-agarose (Qiagen). Tubulin (Cytoskeleton) was pre-cleared by centrifugation at 90000 × $g$ for 10 min and then induced to polymerize by addition of 5% glycerol and incubation at 37°C for 1 hr. Assembled MTs were diluted to 1 mg/ml tubulin, and then added to purified 6XHis-GFP-NuMA-TIP. The 6XHis-GFP-NuMA-TIP was also precleared, but aggregates still precipitated out in the buffer conditions used. We used concentrations of GFP-NuMA-TIP ranging from 300 nM down to 30 nM, but the actual concentrations in solution might be lower due to the aggregation of some of the protein. After 2 min, 1 µM nocodazole was added to promote MT depolymerization. After 1 min, reactions were fixed with 10 volumes of 1% glutaraldehyde in BRB80 for 10 min at 37°C, and then further diluted with PBS. Fixed mixtures were spun onto poly-L-lysine coated coverslips at 25,000 × $g$ for 30 min in a swinging bucket rotor. Coverslips were then fixed and stained for β-tubulin (Sigma).

## X-gal barrier assay

Embryos were isolated from pregnant dams at day e18.5 and incubated at 30°C overnight in X-gal solution (1.3 mM $MgCl_2$, 100 mM $NaPO_4$, 3 mM $K3Fe(CN)_6$, 0.01% Na deoxycholate, 0.2% NP-40, 1 mg/mL X-gal [Invitrogen], pH 4.5).

## Spindle orientation analysis

Spindle orientation in cultured cells was measured by determining the angle between the two spindle poles and the center of the cortical NuMA crescent (as illustrated in *Figure 3C*). Spindle orientation in both interfollicular basal cells and hair follicle matrix cells was measured by determining the angle between a line bisecting both anaphase chromosomes (identified using pHH3 staining) and the underlying basement membrane (β4-integrin staining).

## Statistical analysis

Student's t-tests were used for all statistical analyses, with the exception of the radial histograms of spindle orientation that were analyzed using the Kolmogorov–Smirnov test.

## Acknowledgements

We thank Don Cleveland and Elaine Fuchs for mouse strains, Julie Underwood for expert care of the mice, as well as members of the Lechler Lab for feedback on the manuscript. This work was supported by NIH grants R01AR055926 and R01GM111336 to TL, and by a predoctoral fellowship to AM from NSF.

## Additional information

### Funding

| Funder | Grant reference number | Author |
| --- | --- | --- |
| National Institutes of Health | R01AR067203 | Terry Lechler |
| National Institutes of Health | R01GM111336 | Terry Lechler |
| National Science Foundation | | Andrew Muroyama |

The funders had no role in study design, data collection and interpretation, or the decision to submit the work for publication.

## Author contributions

LS, TL, Conception and design, Acquisition of data, Analysis and interpretation of data, Drafting or revising the article; AM, Drafting or revising the article, Contributed unpublished essential data or reagents

## Author ORCIDs

Terry Lechler, http://orcid.org/0000-0003-3901-7013

## Ethics

Animal experimentation: All mouse studies were performed in accordance with our protocol (A147-15-05) approved by the Institutional Animal Care and Use Committee of Duke University (A147-15-05).

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
