## [Decision Letter]

Thank you for submitting your work entitled "NuMA-microtubule interactions are critical for spindle orientation and the morphogenesis of diverse epidermal structures" for consideration by *eLife*. Your article has been reviewed by three peer reviewers, one of whom, Yukiko Yamashita, is a member of our Board of Reviewing Editors. The evaluation has been overseen by the Reviewing Editor and Janet Rossant as the Senior Editor.

The reviewers have discussed the reviews with one another and the Reviewing Editor has drafted this decision to help you prepare a revised submission.

Summary:

This study examined the role of a previously identified microtubule-binding domain within NuMA in spindle orientation using a very nice mammalian experimental system. The authors shows the requirement of MT binding by NuMA in spindle orientation: NuMA's MT binding domain (NuMA-TIP) binds to "curled MT end" (characteristic to depolymerizing phase) as opposed to growing MT end. The authors convincingly show that recruitment of dynein to the cortex by NuMA is not sufficient for spindle orientation, and instead the correct spindle orientation requires NuMA's MT binding. NuMA mutant that specifically lacks MT binding domain results in spindle orientation defect without compromising any other aspects of spindle (morphology or assembly), providing a unique opportunity to study the role of spindle orientation during development. The authors examined the phenotype of such mutants in the context of epidermis and hair follicle development. Their results clearly demonstrate the critical importance of spindle orientation during development.

Although the mechanism of spindle orientation in the context of development has been studied extensively, there are not many examples where spindle orientation defect solely led to this extent of dramatic phenotype.

Overall, the experiments are executed and interpreted very carefully, meeting the highest standard of cell biology. Yet, the authors further go on to examine the implication of their finding in the context of development. This is a rare example to clearly demonstrate the role of spindle orientation in organogenesis, separating spindle orientation and assembly issues. This manuscript provides fundamental insight for both epithelial and cell biology fields.

Essential revisions:

There are a few important points to be addressed in revision as summarized below.

1) Because it is difficult to show NuMA's localization in tissues (Figure 3 and later) at the resolution shown in Figure 1 and Figure 2 it remains somewhat unclear whether tissue phenotypes indeed stem from NuMA's MT binding domain characterized in Figure 1 and Figure 2 Therefore we recommend using more careful language in explaining the tissue data. Also, if possible, additional experiments can be added to strengthen the points.

1i) Figure 3 and Figure 3—figure supplement 4 show that cortical recruitment of both NuMA and dynactin(/dynein) is retained when NuMA-deltaMTBD is expressed. However, the authors show that this crescent localization no longer aligns with the mitotic spindle. This finding is in contrast with Kotek et al., 2012, which demonstrates that dynein targeted to the plasma membrane is sufficient to direct spindle positioning.

1ii) Figure 5 suggests that K14Cre/NuMA-deltaMTBD mice have an epidermal differentiation defect caused by spindle randomization, which leads to an incomplete barrier. However from the data presented it remains unclear if these defects are direct effects from initial problems in epidermal stratification or if these are secondary/inflammatory effects from the barrier defect. To resolve these possibilities we suggest the authors perform an earlier time-course of epidermal stratification (e14,15,16) within this model to elucidate the primary defects that lead to the phenotype seen by e18.5.

1iii) The authors demonstrate in Figure 6 that the MatrixdeltaMTBD hair follicles lack active BMP signaling in the inner root sheath (IRS). However the differentiation state of these cells remains unclear. If we are to take the authors assumption that these IRS cells from the matrix have not differentiated, then what state are they in? Are they undifferentiated? The authors show BrdU positive cells in Figure 6—figure supplement 3 but is not clear whether those follicles are equivalent in growth stage as the ones analysed in Figure 6. Performing a Brdu pulse at P2 to match the time-points used in main Figure 6 can help to understand whether these cells are in fact stacked into an undifferentiated state.

1iv) Finally, it remains unclear how changes in spindle orientation directly lead to these phenotypes. If cells in the basal layer of the interfollicular epidermis and hair follicle matrix are skewed toward parallel and oblique divisions from the basement membrane, shouldn't the authors see an accumulation of cells in these basal compartments? This is not apparent in any of the data presented. Furthermore, if there is not an increase in basal cell density in these compartments then how do these cells become suprabasal? Additionally, if excess basal cells are able to become suprabasal, then how do they achieve this? Is this a cell autonomous process?

2) Why the MT binding domain is required for spindle orientation is not addressed in this work except to postulate a model where the essential function takes place at the cortex. Thus, the new contributions are a more detailed characterization of NuMA's MT binding activity, the demonstration that this activity is not required for localization of NuMA or Dynein, and the demonstration that this activity is required for spindle orientation, both in vitro and in vivo. On the one hand, it could be argued that the lack of significant mechanistic insight reduces the appeal of the work to non-specialists, however it is also true that our understanding of NuMA's role in spindle orientation has been slow in coming. That said, I believe there are several experimental issues that should be addressed before publication.

2i) A key mechanistic finding is that the microtubule binding activity is not required for localization of NuMA or Dynein to the cortex. This should be supported by quantification and by a negative control (that under the experimental conditions used by the authors, p150glued cortical localization requires NuMA).

2ii) The authors’ model includes simultaneous engagement of both LGN and MTs by NuMA but previous work by Macara's group (Du et al. Current Biology 2002) showed that LGN inhibits NuMA's MT binding activity. The authors should verify that they do not see this competition, which would invalidate their model.

---

## [Author Response]

*1) Because it is difficult to show NuMA's localization in tissues (Figure 3 and later) at the resolution shown in Figure 1 and Figure 2 it remains somewhat unclear whether tissue phenotypes indeed stem from NuMA's MT binding domain characterized in Figure 1 and Figure 2. Therefore we recommend using more careful language in explaining the tissue data. Also, if possible, additional experiments can be added to strengthen the points.*

*1i) Figure 3 and Figure 3—figure supplement 4 show that cortical recruitment of both NuMA and dynactin(/dynein) is retained when NuMA-deltaMTBD is expressed. However, the authors show that this crescent localization no longer aligns with the mitotic spindle. This finding is in contrast with Kotek et al., 2012, which demonstrates that dynein targeted to the plasma membrane is sufficient to direct spindle positioning.*

Yes, we agree and this was a major point of our manuscript. Kotak et al. demonstrated that dynein was sufficient to induce spindle oscillations when targeted in an unpolarized manner to the cell cortex. However, assays for spindle orientation (relative to an internal or external landmark) were not performed. Their data, therefore, demonstrate that cortical dynein can induce forces on spindles, however, they do not show that this is sufficient for robust orientation of the mitotic spindle. Our data demonstrate that it is not. Additionally, it is important to point out that endogenous NuMA was present in these spindle oscillation assays that may have collaborated with dynein/dynactin.

We added the following sentence in the Discussion to emphasize this point. “While a previous study demonstrated that artificially targeting dynein to the cortex was sufficient to induce spindle oscillations (Kotak et al., 2012), our data suggest that cortical dynein is not sufficient to robustly position the spindle and that NuMA-MT interactions are also required (Figure 7).”

*1ii) Figure 5 suggests that K14Cre/NuMA-deltaMTBD mice have an epidermal differentiation defect caused by spindle randomization, which leads to an incomplete barrier. However from the data presented it remains unclear if these defects are direct effects from initial problems in epidermal stratification or if these are secondary/inflammatory effects from the barrier defect. To resolve these possibilities we suggest the authors perform an earlier time-course of epidermal stratification (e14,15,16) within this model to elucidate the primary defects that lead to the phenotype seen by e18.5.*

We have included analysis of differentiation at e16.5 (both stratification and gene recombination begin at e14.5 so this is the earliest time point when we expected to see phenotypes). While not as dramatic as that seen at e18.5, we found many focal regions with disruptions in the transition to differentiation (Figure 5—figure supplement 2). In addition, the phenotypes seen are unlikely to be secondary to inflammatory effects as these do not occur embryonically. As discussed in the initial manuscript, we find that the increase in basal cell proliferation (at e18.5) is likely due to secondary defects as it was not seen at e16.5. This is in contrast to suprabasal proliferation (another sign of defective differentiation), which was present at both e16.5 and e18.5.

*1iii) The authors demonstrate in Figure 6 that the MatrixdeltaMTBD hair follicles lack active BMP signaling in the inner root sheath (IRS). However the differentiation state of these cells remains unclear. If we are to take the authors assumption that these IRS cells from the matrix have not differentiated, then what state are they in? Are they undifferentiated? The authors show BrdU positive cells in Figure 6—figure supplement 3 but is not clear whether those follicles are equivalent in growth stage as the ones analysed in Figure 6. Performing a Brdu pulse at P2 to match the time-points used in main Figure 6 can help to understand whether these cells are in fact stacked into an undifferentiated state.*

The fate of the matrix-derived cells is a very interesting point and one that we remain unclear about. We have looked at proliferation (both by BrdU and by Ki67 staining) and find that it remains restricted to the matrix, see new Figure 6—figure supplement 4). These data suggest that matrix localization is sufficient in this case to restrict proliferative activity. This is likely due to either basement membrane attachment or pro-proliferative cues in the local environment. These data are similar to those seen upon disruption of BMPRIA – inner root sheath (IRS) markers were lost, yet proliferation remained restricted to the matrix. Positive markers for the differentiated cells remain unknown.

We added to the following in the text:

Results: “Although matrix-derived cells in the mutant mice lacked markers for IRS and hair shaft, they appeared to be non-proliferative, as judged by lack of BrdU incorporation (Figure 6—figure supplement 4). “

Discussion: “In both cases, IRS and hair shaft markers are decreased or absent. The matrix-derived cells in the mutant mice described here as well as the BMPRIA mutant are non-proliferative, suggesting that they may be segregated from pro-proliferative signals from the basement membrane and/or dermal papilla.”

*1iv) Finally, it remains unclear how changes in spindle orientation directly lead to these phenotypes. If cells in the basal layer of the interfollicular epidermis and hair follicle matrix are skewed toward parallel and oblique divisions from the basement membrane, shouldn't the authors see an accumulation of cells in these basal compartments? This is not apparent in any of the data presented. Furthermore, if there is not an increase in basal cell density in these compartments then how do these cells become suprabasal? Additionally, if excess basal cells are able to become suprabasal, then how do they achieve this? Is this a cell autonomous process?*

This is a very interesting question and one that we have attempted to address by examining the density of basal cells in the mutant epidermis. We found that there was a 6% increase in density (102+/-2 cells/mm in the control and 108+/-3 cells/mm in the mutant). This result did not reach statistical significance and we have therefore not included it in the text. We should note that knockdown of LGN in vivo caused almost all spindles to be oriented parallel to the basement membrane resulting in symmetric divisions (Williams et al., Nature 2011) and this resulted in only a 17% increase in basal cell density. As that mutation was both more severe and was induced earlier than ours, it is not surprising that a more dramatic effect was not seen. While we do not have data to directly address this, we feel it is likely that cells are lost from the basal layer either through an active delamination or through extrusion. However, this appears insufficient for proper differentiation.

*2) Why the MT binding domain is required for spindle orientation is not addressed in this work except to postulate a model where the essential function takes place at the cortex. Thus, the new contributions are a more detailed characterization of NuMA's MT binding activity, the demonstration that this activity is not required for localization of NuMA or Dynein, and the demonstration that this activity is required for spindle orientation, both* in vitro *and* in vivo*. On the one hand, it could be argued that the lack of significant mechanistic insight reduces the appeal of the work to non-specialists, however it is also true that our understanding of NuMA's role in spindle orientation has been slow in coming. That said, I believe there are several experimental issues that should be addressed before publication.*

*2i) A key mechanistic finding is that the microtubule binding activity is not required for localization of NuMA or Dynein to the cortex. This should be supported by quantification and by a negative control (that under the experimental conditions used by the authors, p150glued cortical localization requires NuMA).*

We have included a quantitation (subheading “NuMA’s MTBD is required for epidermal spindle orientation and differentiation.”). We find that 100% of cells with detectable cortical NuMA have p150-glued co-localized at this site. This is true for both the control cells and the NuMA-MTBD deleted cells. Although we do not include the negative control in this paper, we recently published that upon knockdown of NuMA, over 95% of cells lose detectable p150-glued cortical localization (consistent with findings from a number of other investigators).

We add the following to the Results section:

“This was true in 100% of cells examined (n=50 from two experiments). In contrast, we recently published that knockdown of NuMA resulted in an almost complete loss of cortical dynactin (Seldin et al., 2013).”

*2ii) The authors’ model includes simultaneous engagement of both LGN and MTs by NuMA but previous work by Macara's group (Du* et al. *Current Biology 2002) showed that LGN inhibits NuMA's MT binding activity. The authors should verify that they do not see this competition, which would invalidate their model.*

We have no reason to doubt the mutual exclusivity of binding that was reported previously. In fact, the finding that the NuMA-TIP binding domain includes the LGN-binding site further suggests that these will compete.

That said, it is important to note that those experiments were performed on monomeric fragments of NuMA. Full-length NuMA is an obligate dimer that also multimerizes (dodecamer when purified, Harborth et al. EMBO 1999). Therefore it is most likely that LGN and MTs can simultaneously interact with the complete complex, though not with isolated domains.